# 222-Nanometer Far-UVC Exposure Results in DNA Damage and Transcriptional Changes to Mammalian Cells

**DOI:** 10.3390/ijms23169112

**Published:** 2022-08-14

**Authors:** Qunxiang Ong, Winson Wee, Joshua Dela Cruz, Jin Wah Ronnie Teo, Weiping Han

**Affiliations:** 1Institute of Molecular and Cell Biology, Agency for Science, Technology and Research (A*STAR), 11 Biopolis Way, #02-02, Helios, Singapore 138667, Singapore; 2Singapore Institute of Manufacturing Technology (SIMTech), Agency for Science, Technology and Research (A*STAR), 2 Fusionopolis Way, #08-04, Innovis, Singapore 138634, Singapore

**Keywords:** far UVC, UVC LED, senescence, DNA damage, signaling pathways

## Abstract

Ultraviolet (UV) germicidal tools have recently gained attention as a disinfection strategy against the COVID-19 pandemic, but the safety profile arising from their exposure has been controversial and impeded larger-scale implementation. We compare the emerging 222-nanometer far UVC and 277-nanometer UVC LED disinfection modules with the traditional UVC mercury lamp emitting at 254 nm to understand their effects on human retinal cell line ARPE-19 and HEK-A keratinocytes. Cells illuminated with 222-nanometer far UVC survived, while those treated with 254-nanometer and 277-nanometer wavelengths underwent apoptosis via the JNK/ATF2 pathway. However, cells exposed to 222-nanometer far UVC presented the highest degree of DNA damage as evidenced by yH2AX staining. Globally, these cells displayed transcriptional changes in cell-cycle and senescence pathways. Thus, the introduction of 222-nanometer far UVC lamps for disinfection purposes should be carefully considered and designed with the inherent dangers involved.

## 1. Introduction

It is well established that mammalian cells sense ultraviolet (UV) irradiation and mount a series of elaborate responses, also known as the UV response, which targets multiple signaling pathways [1]. These could eventually lead to apoptosis and local DNA damage [2]. The long-term negative consequences of UV irradiation include erythema [3], skin cancer [4], and damage to the cornea [5] and lens [6,7].

Traditionally, we have focused our attention on UVA (330–400 nm) and UVB (290–330 nm) given that these are the UV wavelengths to which we are typically exposed on the Earth’s surface. UVC (200–280 nm) is believed to be absorbed by the atmospheric ozone and does not reach the earth’s surface [8]. However, with the increasing use of man-made UV germicidal irradiation tools, it is likely that more human exposure to UVC could occur. This is especially relevant during the COVID-19 pandemic, during which UVC light manufacturers have introduced several UVC devices intended for home usage, and safety precautions have been largely neglected in this process.

Various studies have documented the potential dangers of UVC mercury lamps (254 nm), as a consequence of which temporary eye and skin damage can be observed upon prolonged direct exposure to UVC [9]. The irradiation of mammalian cells with UVC has been known to induce DNA damage in the form of thymine dimers and 6,4-photoproducts [1,10], the activation of MAPK signaling pathways [11,12], and potentially cell-cycle arrest and apoptotic pathways [2]. Recent developments in UV technologies have seen UVC LED, mainly in the wavelengths of ~270–280 nm [13], and 222-nanometer far UVC lamps [14,15] emerging as viable alternatives to the traditional UVC mercury lamp. However, their safety profiles on mammalian cells have been much less thoroughly documented, with that of the 222-nanometer wavelength being controversial. Promisingly, the 222-nanometer far UVC lamp has been touted as a safe solution, in which no pyrimidine dimer formation was observed in human and mouse skin models, and the cell viability of layered cell sheets is retained after irradiation with 222-nanometer far UVC [14]. However, some reports emerged suggesting that 222-nanometer sources may not be as safe, since they are capable of inducing both erythema and cyclopyrimidine dimer (CPD) formation in human skin [16].

Here, we sought to compare the different UVC wavelengths in their effects on cell viability and DNA damage. We found that human retinal cells, ARPE-19 cells, have decreased viability and lower growth rates when exposed to 222-nanometer far UVC compared to control cells. They exhibited cell viability, with the JNK/ATF2 pathway remaining suppressed compared to 254-nanometer and 277-nanometer treatment. However, they had the highest DNA damage, as evidenced by yH2AX staining. The RNA sequencing of cells exposed to 222-nanometer far UVC showed that the cell cycle machinery was disrupted, with DNA damage pathways and senescence pathways activated.

## 2. Results

### 2.1. UVC-Induced Cell Viability Reduction and Apoptotic Processes

We first examined the viability of ARPE19 cells exposed to different UVC wavelengths via a sulforhodamine B (SRB) assay at 2, 5, 9, and 14 days after UVC illumination (Figure 1A). The 254-nanometer and 277-nanometer illumination resulted in low cell viability throughout, while the 222-nanometer far UVC caused the cell viability to be significantly lower than the non-illuminated control cells at days 9 and 14. The UVC-induced lowering of cell viability is dose-dependent, as evidenced by the SRB assay taken 9 days after the UVC illumination (Figure 1B).

To gain a comprehensive understanding of the cellular growth profile, we utilized the xCELLigence platform to track the dynamic changes in cell viability per condition. In agreement with the results from the SRB assays, the 222-nanometer UVC resulted in a significant reduction in cell numbers, while the 254-nanometer and 277-nanometer UVC saw no viable cell growth within the 6-day experimental period (Figure 1C). The 222-nanometer-induced reduction in cell viability was dose-dependent, as demonstrated in Appendix A.

We then carried out trypan-blue staining to track the cell death of the ARPE19 1 day after the varied UVC wavelength illumination (Figure 1D), where 254-nanometer and 277-nanometer UVC posted a high percentage of trypan-blue-stained cells at 60.5% and 41.2% respectively. This is corroborated by Western blot analysis, which saw caspase-3 activation prominently in 254-nanometer- and 277-nanometer-lit cells (Figure 1E).

### 2.2. Differential Signaling Pathway Activation under Varied UVC Wavelengths

It is well established that mitogen-activated protein kinases (MAPKs) mediate the UV response, which determine cell fate [11]. These include the extracellular signal-regulated kinases (ERKs), the c-Jun NH2-terminal kinases (JNKs), and the p38 kinases (Figure 2A). We hypothesize that MAPKs could be regulated differently by UVC wavelengths, which result in diverse cellular outcomes. Therefore, we subjected ARPE-19 cells to 20 and 60 min of the respective UVC wavelengths and monitored the activation of the MAPKs. Figure 2B shows that while the ERK and p38 pathways were activated under all three UVC wavelengths, phosphorylation of JNK/ATF2 pathway was not observed under 222-nanometer illumination.

It has also been shown that the PERK/eIF2a axis is crucial for mediating endoplasmic-reticulum-stress signaling upon UV irradiation [17] (Figure 2C). We noticed that the phosphorylation events of PERK and eIF2a did not occur in the 222-nanometer-lit cells a compared to the 254-nanometer and 277-nanometer-illuminated cells (Figure 2D). Given that both JNK and PERK/eIF2a pathway activation have been linked to apoptosis events, it could be inferred that the lack of activation of these stress-signaling pathways in 222-nm-lit cells could be responsible for sustaining ARPE19 cell survival [18,19].

### 2.3. DNA-Associated Damage or Associated Repair Mechanisms Were Observed in 222-Nanometer-Lit Cells

Upon determining the lack of apoptotic mechanisms in the 222-nm-lit cells, we next investigated whether physical damages are sustained in these cells compared to other wavelengths. Using Coomassie staining, we observed that ARPE19 cells subjected to 60 min of 222-nanometer treatment had a reduction in intensity of several protein bands (Appendix A). The performance of 4-hydroxyonenal (4-HNE) staining further revealed increases in oxidative stress due to increased lipid peroxidation chain reaction (Appendix A).

UVC is also well known to cause DNA damage to cells, in which the formation of cyclobutane dimers and (6, 4)-photoproducts has been well characterized [2,10]. In recent years, yH2AX has emerged as a reliable marker for double-strand breaks and their repair [20]. We hypothesized that the DNA in mammalian cells could potentially undergo damage under the different UVC wavelengths, and that the subsequent DNA-associated damage and repair mechanisms could be visualized with yH2AX staining. Using fluorescence microscopy, we observed that 222-nanometer far UVC resulted in a higher percentage of yH2AX-positive cells compared to the 254-nanometer and 277-nanometer UVC wavelength illumination. (Figure 3A,B) The number of foci in the 222-nanometer-illuminated cells remained high 48 h after the initial illumination, indicating the persistence of these foci (Figure 3C).

In comparison, 222-nanometer far UVC did not elicit any cyclobutane dimer formation while the formation of these dimers is most apparent due to 254-nanometer and 277-nanometer illumination (Figure 3D,E). This indicates that the DNA damages or repair mechanisms elicited from the 222-nanometer far UVC did not arise from cyclobutane dimer formation and were potentially derived from other forms of stress insults. These observations were congruent in the HEKA keratinocytes (Appendix A), which indicates that similar mechanisms could be at play in different mammalian cells.

### 2.4. Perturbation of Transcription and Cellular Signaling Events Was Observed in 222-Nanometer-Lit Cells

Given that 222-nanometer far UVC illumination causes damage to the DNA of ARPE-19 cells, we next investigated whether the cellular transcriptional programs and key signaling modalities are restructured. We first examined the transcriptional changes via qRT-PCR of the ARPE-19 cells 1 day after 60 min of 222-nanometer far-UVC illumination and focused on tumor-suppressor genes (Rb1 and P53), cancer-related genes (SNAI1, E2F1, E2F2, and CDK4), and stress-related genes (BMP4, NRF2, p21, IL6, and HIF1A) (Figure 4A). Notably, there was a significant upregulation of the SNAI1 and p21 genes, and key genes that are downregulated include BMP4, p53, E2F2, and E2F1 genes. These are corroborated by Western blot analyses (Figure 4B,C), where both the tumor-suppressor genes Rb1 and p53 saw a reduction in band intensity in 1–3 days after 222-nanometer far-UVC illumination, whilst phosphor-ERK and its downstream signaling target SNAI1 posted an increase 1–3 days after 222-nanometer exposure. Interestingly, the protein levels of the tumor-suppressor genes and the extent of the phosphor-ERK recovered to their original levels 6 days after the exposure.

### 2.5. Analysis of Global RNA Transcripts One Week after 222-Nanometer Exposure

In order to investigate whether the change in transcriptional activity was a transient phenomenon, we collected the RNA from the cells exposed to no light or 222-nanometer far-UVC illumination for a week and performed RNA sequencing on both sets. Figure 5A plots the top differentially expressed genes derived from the RNA-sequencing results and Appendix A depicts the principal component analysis of the triplicates used in the analysis. We then identified the statistically significant genes (*p* < 0.05, *n* = 2361) and analyzed this subset using ingenuity pathway analysis (IPA). Figure 5B reveals the top ingenuity canonical pathways, among which the top upregulated pathways included DNA damage, checkpoint control, and regulation, while the top downregulated pathways included cell-cycle and replication pathways. The same trends were reflected in the Diseases and Disorders set, where a large percentage (~90%) of the genes were implicated in the larger category of Cancer and Organismal Injury and Abnormalities (Figure 5C). Out of these pathways, the top cellular functions covered cell death and survival and cell cycle. We then summarized these IPA analysis findings in the summary figure (Figure 5D), which shows the top pathways implicated, including the formation of gamma H2AX foci and the senescence of cells being upregulated, whilst the cell-growth-related pathways, such as entry into interphase and colony formation, were affected. Appendix A contain further information regarding the top differentially expressed genes, the top regulator pathway, and the upstream regulator networks, as informed by the IPA analysis.

In addition, we also conducted a Gene-set enrichment analysis (GSEA) based on the detection of 20,358 genes in the RNA sequencing profiles, and found that the results corroborated with the IPA analysis. Figure 5E shows the two representative key pathways up/downregulated in the form of DNA damage/Telomere-stress-induced senescence (normalized enrichment score of 1.473) and KEGG Cell Cycle (normalized enrichment score of −1.4747). Further GSEA analysis can be found in Appendix A.

## 3. Discussion

As the COVID-19 pandemic has developed, highlighting the importance of public disinfection tools in coping with COVID-19 and future viruses in the post-COVID world, 222-nanometer far UVC has emerged as a potential safe solution in disinfecting viruses, including SARS-CoV-2, whilst presenting as a safe option that allows human exposure [12,13,14,21,22]. The safety presumption of 222-nanometer far UVC has been largely based on the argument that the 222-nanometer far UVC cannot penetrate the outer, non-living cells of the eyes and skin, and that it does not induce typical UVC-induced DNA lesions in human keratinocyte models or the skin of exposed hairless mice. The former has not yet been validated experimentally and, to date, there has been a lack of long-term studies on human exposure. While the data presented by the latter seem promising, the results call for deeper and further investigations as those conducted previously only covered typical UVC-induced DNA lesions and not general classes of DNA damages.

In this paper, we exposed the skin and eye cells to 20 and 60 min of respective UVC wavelengths and studied the immediate and longer-term effects of UVC damage towards cellular health and viability. In agreement with earlier reports, the 222-nanometer-lit cells survived and retained their ability to grow without the activation of apoptosis, whilst the 254-nanometer and 277-nanometer-lit cells showed decreased viability and had apoptosis activated [14]. We then studied the mechanisms leading to the lack of this activation and found that, potentially, the pJNK/ATF2 and the endoplasmic-reticulum-stress pathways may be involved. However, we found that the 222-nanometer-lit cells had a lower growth rate compared to the unilluminated controls. We then pursued the DNA-damage pathways and saw that while the 222-nanometer far UVC did not incur any thymine dimer formation, unlike the 254-nanometer UVC lamp and 277-nanometer UVC LED, it recorded the highest incidence of γH2AX nuclear foci, indicating DNA damage. This was particularly alarming given that the tumor-suppressor genes, Rb1 and p53, ere downregulated in the 222-nanometer-lit cells one to three days after the exposure. RNA sequencing of the cells exposed to the 222-nanometer illumination was also conducted to find that the cellular survival and cell-cycle pathways were re-sculpted, with the summary analysis showing the upregulation of γH2AX nuclear foci and senescence of cells, corroborating all the previous results.

It is alarming that clinical scientists and researchers have taken the initial optimism regarding the safety profile of 222-nanometer far UVC and begun to pursue clinical trials for performing such illumination on open wounds [23] and in healthy humans [24]. The excitement generated in the field has also seen significant interest in this technology from hospitals and facilities management, which have adopted the tool for open human exposure.

## 4. Materials and Methods

### 4.1. Key Resources Table

The Key Resources Table (Appendix A) includes the source and identifier of antibodies, media, commercial assays, cell lines, oligonucleotides for qPCR, software, and hardware used in the study.

### 4.2. UVC Sources and Irradiance Measurements

To determine the effect of UVC wavelength on human cell lines, three different UVC light sources—222 nm far UVC lamp (Ushio, Tokyo, Japan), 254 nm UVC mercury lamp (Osram), and 277 nm UVC LED (Lextar, Hsinchu, Taiwan, PU35CM1) were used in this study. The detailed method was described in a previous work [25], These UVC light sources were measured using a calibrated spectroradiometer (GL Spectis 4.0 from GL Optic, Würzburg, Germany) with an absolute measurement uncertainty of less than 6%, from 200 to 500 nm. To provide a comparative UVGI efficacy study between these UVC light sources, the radiant intensity of the far UVC and mercury lamp was measured at different distances, while the UVC LED was driven at different constant drive currents to obtain a common UV intensity of 73 µW/cm^2^. The UVC LED, with a beam angle of 120°, was assembled into a 5 × 5 array at a working distance of 12 cm to ensure uniform UV intensity across the surface of the petri dish.

### 4.3. Cell Culture

The ARPE-19 retinal cells were grown in DMEM:F12 (ATCC Manassas, VA, USA, Catalog No. 30-2006) supplemented with 10% Fetal Bovine Serum (FBS), 100 U/mL penicillin and 100 µg/mL streptomycin (Sigma-Aldrich, St. Louis, MO, USA). The HEK-A keratinocytes (ATCC, Manassas, VA, USA, PCS-200-011) were cultured in Dermal Cell Basal Medium supplemented with the keratinocyte growth kit (ATCC, Manassas, VA, USA), 100 U/mL penicillin and 100 ug/mL streptomycin (Sigma-Aldrich, St. Louis, MO, USA).

### 4.4. Cell Viability Assays

To measure the effect of different UVC illumination on cell viability, the cells were exposed to 0.073 µW/cm^2^ of respective wavelengths for an hour and then analyzed after the defined recovery time. These were performed with the xCELLigence platform (Roche, Basel, Switzerland), and also through sulforhodamine B (SRB) assay.

### 4.5. XCELLigence Platform

The cells were irradiated in 100-millimeter petri dishes and allowed to recover overnight before being seeded into the appropriate 96-well xCELLigence plates at a density of 1000 cells per well. Following seeding, the cells were monitored every 10 min by the xCELLigence system (Roche, Basel, Switzerland) for proliferation, attachment, and spreading. The impedance detection was performed for a total of 6 days. Real-Time Cell Analysis 2.0 (Roche, Basel, Switzerland) software was used to analyze the data.

### 4.6. SRB Assay

The viability of each condition was determined using a colorimetric SRB assay [26]. The cells were irradiated in 100-millimeter petri dishes and allowed to recover overnight before being seeded into 96-well tissue-culture plates at a density of 1000 cells per well. For the defined number of days following irradiation, 100 µL cold 10% trichloroacetic acid were added and allowed to incubate at 4 °C for 1 h. The plates were then washed with distilled water and dried in 37 °C for 30 min, before 80 µL of SRB dye was added to each well for 5 min. The plates were then washed with 1% acetic acid and dried in 37 °C for 30 min; the dye was then resolubilized in 10 mM Tris at room temperature for an hour. The plates were then recorded for the absorbance at a wavelength of 510 nm, with the values expressed in arbitrary units. All SRB assay experiments were performed in triplicate.

### 4.7. Immunofluorescence Experiments

To assess whether various levels of UVC illumination caused different types of DNA damage to ARPE-19 and HEK-A cells, immunostaining was performed to detect the presence of γH2AX and thymine dimers in these cells. Briefly, 2 × 10^5^ cells were plated in each petri dish one day before the experiment. After the described duration of UVC illumination, the cells were washed with PBS and fixed with 4% paraformaldehyde at room temperature for 15 min and washed with PBS before being labelled with anti-phospho-Histone H2A.X (Ser139) antibody (Sigma Aldrich, St. Louis, MO, USA, Cat#:05-636) 1:500 or anti-Thymine dimer antibody (Novus Biological, Cambridge, UK, Cat#: NB600-1141) 1:250 in PBS containing 2% bovine serum albumin (BSA) and 0.1% TBS-T. Cells were then washed with PBS and labeled with goat anti-rabbit Alexa Fluor-568 (Life Technologies, Grand Island, NY, USA) in PBS containing 2% BSA at room temperature for an hour with gentle shaking. Following washing with PBS, the cells were stained with DAPI and observed with the 40× objective of Nikon Ti-2 TIRF microscope. Post-experiment analysis of the levels of γH2AX and thymine dimers was performed by calculating the average intensity of the red channel minus the image background I_bg_. I_bg_ is defined as the average intensity of an image from blank areas where no DAPI signal is observed.

### 4.8. Western Blot Experiments

All samples for Western blots were lysed in NP40 buffer supplemented with phosphatase and protease inhibitors. Approximately 10 µg of protein sample were loaded for Western blot experiments. The lysates were then subjected to SDS gel electrophoresis before being transferred to nitrocellulose membranes using iBlot2 (Life Technologies, Grand Island, NY, USA), blocked with 5% BSA in TBS with 0.1% Tween-20 and incubated with primary antibodies. Membranes were then incubated with rabbit-IRDye 800 CW secondary antibodies and imaged on an Odyssey CLx (LI-COR Biosciences, Lincoln, NE, USA).

### 4.9. RNA Extraction and Analysis

RNA was isolated from ARPE-19 cells by Trizol isolation and genomic DNA was subsequently removed with DNAseI (Invitrogen, Carlsbad, CA, USA). The cDNA was then generated from 1 µg of RNA using RevertAid First Strand cDNA Synthesis Kit (Thermofisher, Waltham, MA, USA) as per manufacturer’s recommendations. Quantitative PCR was then carried out with PowerUp SYBR Green Master Mix (Life Technologies, Grand Island, NY, USA) and the analysis was performed using the Quantstudio 5 software.

### 4.10. RNA Sequencing

Total RNA from control and 222-nm-treated ARPE-19 cells was isolated and 1 µg of RNA per sample was used as the input material. Next-generation-sequencing library preparations were constructed according to the manufacturer’s protocol. The poly(A) mRNA isolation was performed using poly(A) mRNA magnetic isolation module or rRNA-removal kit. The mRNA fragmentation and priming were performed using first-strand synthesis reaction buffer and random primers. First-strand cDNA was synthesized using ProtoScript II reverse transcriptase and the second-strand cDNA was synthesized using second-strand synthesis enzyme mix. The purified double-stranded cDNA by beads was then treated with End Prep Enzyme Mix to repair both ends and add a dA-tailing in one reaction, followed by a T-A ligation to add adaptors to both ends. Size selection of adaptor-ligated DNA was then performed using beads, and fragments of ~400 bp (with the approximate insert size of 300 bp) were recovered. Each sample was then amplified by PCR using P5 and P7 primers, with both primers carrying sequences that can anneal with flow cell to perform bridge PCR and P5/ P7 primer carrying index, allowing multiplexing. The PCR products were cleaned up using beads, validated using an Qsep100 (Bioptic, Taiwan, China), and quantified by Qubit3.0 Fluorometer (Invitrogen, Carlsbad, CA, USA). Then libraries with different indices were multiplexed and loaded on an Illumina Novaseq instrument according to manufacturer’s instructions (Illumina, San Diego, CA, USA). Sequencing was carried out using a 2 × 150 paired-end (PE) configuration.

### 4.11. Quantification of Gene Expression

Quality of the sequences was assessed with FastQC v0.11.5 [27]. No further data filtering and trimming were performed. The paired FASTQ files were aligned to hg19 reference genome using GENCODE v.36 gene annotations and STAR v2.6.0a splice aware aligner. Unique transcripts were then assembled from merged alignment files, and abundance of the transcripts are generated by featureCounts. Differentially expressed genes (DEGs) were determined with DESeq2.

### 4.12. Pathway Analyses

In total, 2359 genes were differentially expressed between the illuminated and control group (adjusted Benjamini-Hochberg *p*-value < 0.05), with a total of 1319 downregulated and 1040 upregulated genes. These genes were used for ingenuity pathway analysis (Qiagen, Hilden, Germany). The gene-set-enrichment Analysis was performed with the software version 4.0.3.

### 4.13. Quantification and Statistical Analyses

The number of replicates (*n*) is indicated in the respective figure legends. For all statistical tests, significance was indicated where * *p* < 0.05, ** *p* < 0.01, *** *p* < 0.001, and **** *p* < 0.0001. All statistical analyses were performed with the GraphPad Prism Software 8.4.3 (GraphPad Software Inc., La Jolla, CA, USA) and all values are expressed as means +/− standard deviation.

## Figures and Tables

**Figure 1 ijms-23-09112-f001:**
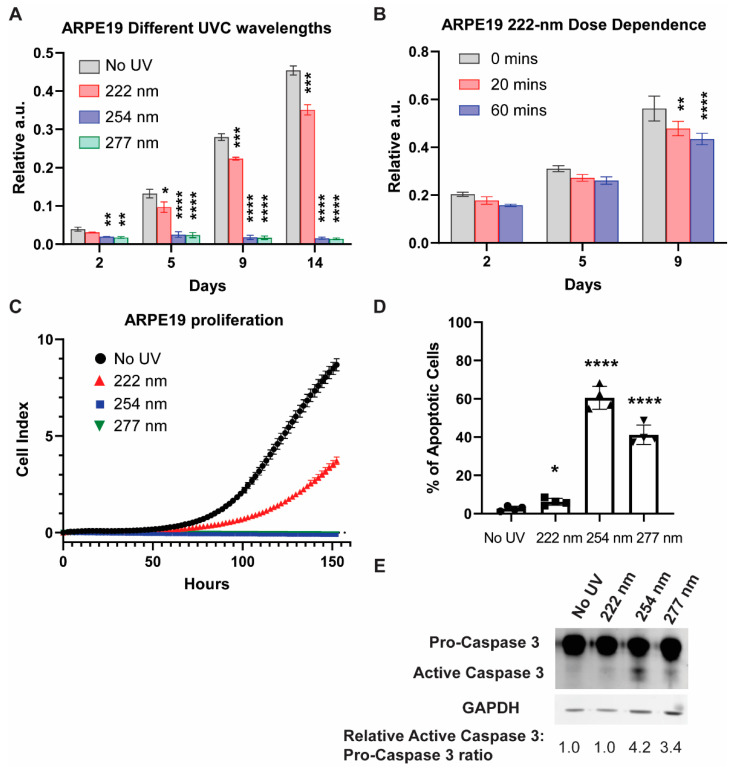
254-nanometer and 277-nanometer UVC sources result in cell death driven by apoptosis, while 222-nanometer far UVC results in decreased cell viability compared to control. (**A**) SRB assay for ARPE-19 cells exposed to either no UVC or 60 min of respective UVC wavelengths and then incubated for the indicated number of days. Values are reported as mean +/− S.D. from *n* = 3 experiments. (**B**) SRB assay for ARPE-19 cells exposed to either 0, 20 or 60 min of 222-nanometer far UVC and then incubated for the indicated number of days. Values are reported as mean +/− S.D. from *n* = 3 experiments. (**C**) Dynamic monitoring of cell numbers through xCelligence platform. Values are reported as mean +/− SD from *n* = 8 replicates. (**D**) Trypan blue staining assay performed on ARPE-19 cells exposed to either no UVC or 60 min of respective UVC wavelengths and then incubated for 1 day. Values are reported as mean +/− SD from *n* = 4 experiments. (**E**) Representative Western blot analysis of pro-caspase 3 versus cleaved caspase 3 after either no UVC or 60 min of respective UVC wavelengths. GAPDH immunoblotting was used as an internal control. Where applicable, Student’s *t*-test is performed and significance is represented as * *p* < 0.05, ** *p* < 0.01, *** *p* < 0.001, and **** *p* < 0.0001.

**Figure 2 ijms-23-09112-f002:**
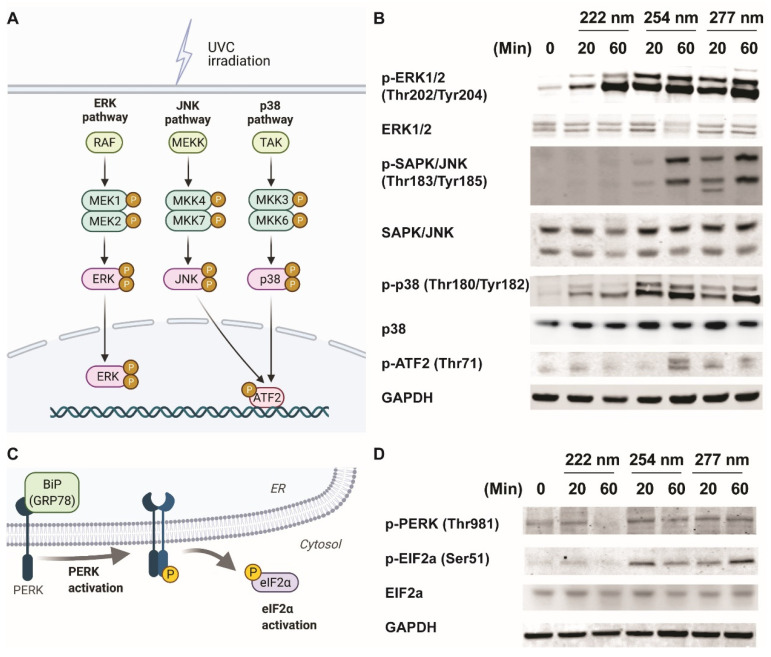
Differential activation of MAPK and ER-stress signaling pathways results in the differential activation of apoptotic pathways. (**A**) Schematic indicating the classical MAPK signaling pathways studied due to classical 254-nanometer UVC irradiation. (**B**) Representative Western blots of phosphor-ERK (Thr202, Tyr204), total ERK1/2, phosphor-JNK/SAPK (Thr183, Tyr185), total JNK1/2, phosphor-p38 (Thr180, Tyr182), p38, phosphor-ATF2 (Thr69), and GAPDH from ARPE-19 cells exposed to 0, 20, and 60 min of respective UVC wavelengths. (**C**) Schematic indicating the proteins involved in endoplasmic-reticulum-induced stress pathway. (**D**) Representative Western blots of phosphor-PERK (Thr980), total ERK1/2, phosphor-EIF2a (Ser49), EIF2a, and GAPDH from ARPE-19 cells exposed to 0, 20, and 60 min of respective UVC wavelengths.

**Figure 3 ijms-23-09112-f003:**
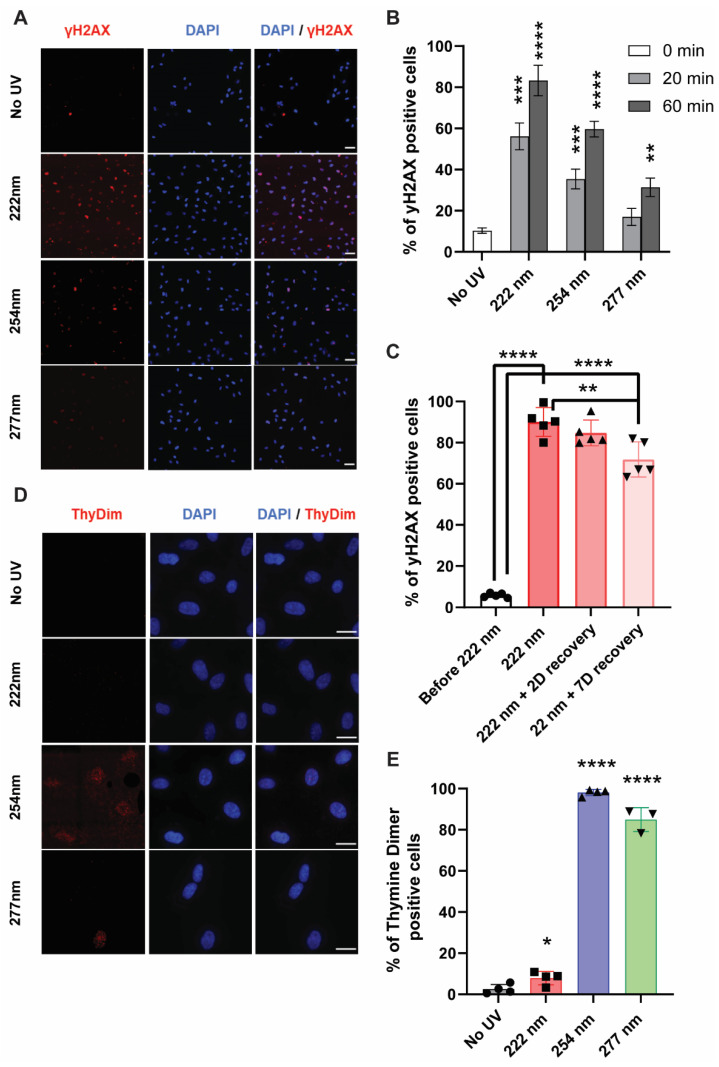
DNA-associated damage or associated repair mechanisms were observed in 222-nanometer-lit cells. (**A**) γH2AX staining in ARPE19 cells upon 60 min of respective UVC irradiation. Scale bars = 50 µm. (**B**) Quantification of the extent of γH2AX activation in the ARPE19 cells upon 0, 20, and 60 min of UVC illumination. Values are reported as mean +/− SD from *n* = 3 experiments. (**C**) Quantification of the extent of γH2AX activation in the ARPE19 cells exposed to 60 min of 222-nanometer far-UVC illumination and then recovered 0, 2, and 7 days after the far-UVC treatment. Values are reported as mean +/− SD from *n* = 5 experiments. (**D**) Thymine dimer staining in ARPE19 cells upon 60 min of respective UVC irradiation. Scale bars = 20 µm. (**E**) Quantification of the extent of thymine dimer formation in the ARPE19 cells after 60 min of UVC illumination. Values are reported as mean +/− SD from *n* = 3–4 experiments. Where applicable, Student’s *t*-test is performed and significance is represented as * *p* < 0.05, ** *p* < 0.01, *** *p* < 0.001, and **** *p* < 0.0001.

**Figure 4 ijms-23-09112-f004:**
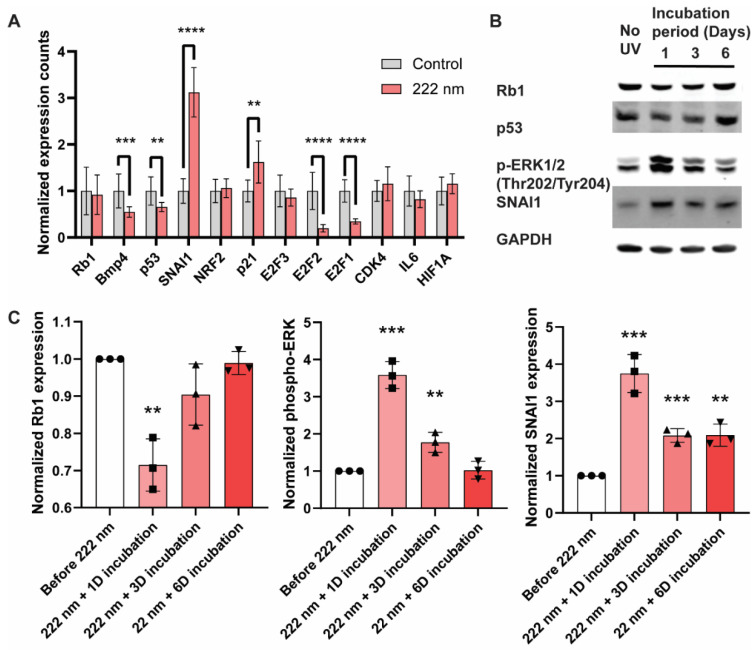
Perturbation of transcription and cellular signaling events was observed in 222-nanometer-lit cells. RT-PCR and Western blot analysis of ARPE-19 cells after 222-nanometer far-UVC illumination. (**A**) RT-PCR analysis of selected genes 1 day after 60 min of 222-nanometer far-UVC illumination. (**B**) Representative Western blots of Rb1, p53, phosphor-ERK (Thr202, Tyr204), SNAI1, and GAPDH from ARPE-19 cells 1, 3, and 6 days after being exposed to 60 min of 222-nanometer far UVC illumination. (**C**) Quantification of the band intensity for Rb1, p53, phosphor-ERK (Thr202, Tyr204) and SNAI1 relative to internal-control GAPDH in the ARPE19 cells 1, 3, and 6 days after 60 min of UVC illumination. Values are reported as mean +/− SD from *n* = 3 experiments. Where applicable, Student’s *t*-test is performed and significance is represented as ** *p* < 0.01, *** *p* < 0.001, and **** *p* < 0.0001.

**Figure 5 ijms-23-09112-f005:**
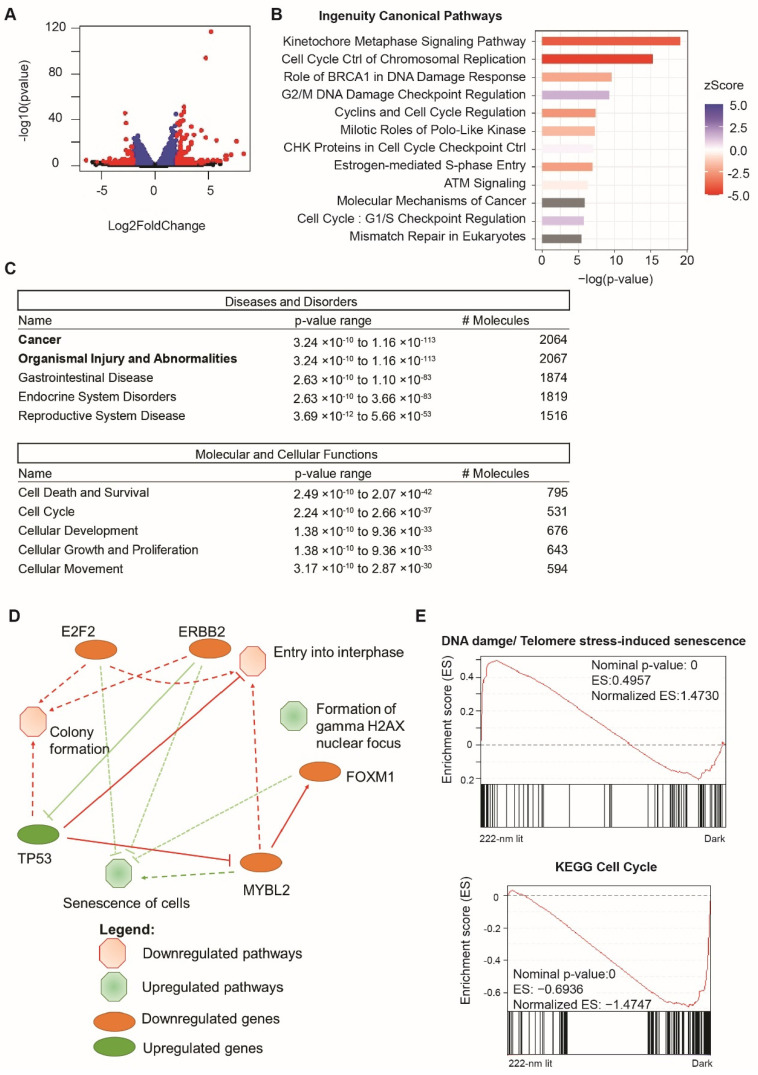
Analysis of global RNA transcripts one week after 222-nanometer exposure (**A**) Highlighted in red are genes that are either upregulated with log2FoldChange at more than 2 and *p* < 0.05 (*n* = 140) or downregulated with log2FoldChange at less than 2 and *p* < 0.05 (*n* = 73). (**B**) Top ingenuity canonical pathways involved due to 222-nanometer illumination with ingenuity pathway analysis performed on 2361 statistically significant differentially expressed genes at *p* < 0.05. A positive activation zScore reflects upregulation of pathway in 222-nanometer illuminated ARPE-19 cells, while a negative activation zScore reflects the opposite. (**C**) Top diseases and disorders and molecular and cellular function categories derived from ingenuity pathway analysis out of 2361 statistically significant differentially expressed genes. (**D**) Ingenuity pathway analysis graphical summary providing a quick overview of the major biological themes and how they relate to each other. (**E**) Gene-set enrichment analysis of all 20358 genes detected in the RNA sequencing found upregulation of DNA damage and telomere-stress induced senescence pathway and a reduction in cell-cycle genes in 222-nanometer-illuminated ARPE-19 cells.

## Data Availability

The data presented in this study are available on request from the corresponding author. The RNA-sequencing data will be made available on GeoDataSets once accepted.

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
