# Peer review of "222-Nanometer Far-UVC Exposure Results in DNA Damage and Transcriptional Changes to Mammalian Cells"

_ijms, 2022, doi:10.3390/ijms23169112_

Round 1

Reviewer 1 Report

The topic of the manuscript is essential nowadays. The Authors presented a sufficient amount of results using several techniques. However, the study has certain serious shortcomings and in general, the manuscript was prepared carelessly.

SPECIFIC COMMENTS:

1.       Why the cells were analyzed at 2, 5, 9, and 14 days after UVC illumination? Please justify the choice of times in which the experiments were performed.

2.       There is a lack of description of statistical data analysis and an explanation of statistical determinations on graphs.

3.       LINE 92-94 “This is corroborated by Western blot analysis which saw caspase-3 activation prominently in 254- 93 nm- and 277-nm-lit cells”. In my opinion, the sentence is not supported by the presented results. There was higher level of GAPDH in the sample “277-nm” and thus the density measurements should be made to draw reliable conclusions.

4.       Figure 3: the background of the images (3A:222nm, 3D: 254nm) in the red channel should be unified or all compared images. Did the authors consider the background intensity during the measurement/analysis of images?

5.       How many microscopic fields and how many cells were analyzed during the immunofluorescence analysis?

6.       Materials and methods: the table is too detailed. I suggest, that it should be presented as supplementary material.

7.       The Authors included the results in the Introduction section. It is not a mistake but I suggest the fragment line 51-line 58 be rewritten.

8.       line 28 – „eye damages” – the term should be explained by including examples.

9.       The manuscript has many editorial errors, e.g. the caption of each figure requires a complete rework.

10.   Reference list was not prepared according to the Journal guidelines.

Author Response

The topic of the manuscript is essential nowadays. The Authors presented a sufficient amount of results using several techniques. However, the study has certain serious shortcomings and in general, the manuscript was prepared carelessly.

Response: We thank the reviewer for the comments and his reminder. In the revised manuscript, we have changed the format carefully, especially in relation to point 6, 9 and 10.

SPECIFIC COMMENTS:

  1. Why the cells were analyzed at 2, 5, 9, and 14 days after UVC illumination? Please justify the choice of times in which the experiments were performed.

Response: The cells are analyzed at 2, 5, 9 and 14 days after UVC illumination as per previous studies in the field of cancer or cell viability where the duration has been classically up to 1-2 weeks. We chose these timepoints to include at least 7 replication cycles as it has been well studied that classical UVC wavelengths would induce apoptosis and induce DNA damages, and such a timepoint will allow enough timeframe for changes in cell viability to be observed.

  1. There is a lack of description of statistical data analysis and an explanation of statistical determinations on graphs.

Response: We note the Reviewer’s comments and have added a section on “Quantification and statistical analysis” in the revised manuscript. The changes are as follows:” The number of replicates (n) are indicated in the respective figure legends. For all statistical tests, significance was indicated where *P < 0.05, **P < 0.01, ***P < 0.001 and ****P < 0.0001. All statistical analyses were performed with the GraphPad Prism Sotware 8.4.3 (GraphPad Software Inc., 377 La Jolla, CA, USA) and all values are expressed as means +/- standard deviation.”

  1. LINE 92-94 “This is corroborated by Western blot analysis which saw caspase-3 activation prominently in 254- 93 nm- and 277-nm-lit cells”. In my opinion, the sentence is not supported by the presented results. There was higher level of GAPDH in the sample “277-nm” and thus the density measurements should be made to draw reliable conclusions.

Response: In the revised manuscript Figure 1E, we have quantified the ratio of active caspase-3 to procaspase-3 (via densitometric analysis) for the different cells and identified that the ratio is increased in 254-nm and 277-nm lit cells at 4.2 times and 3.4 times of the ones without UV illumination respectively. On the other hand, no substantial change of the ratio (1.0) is seen for 222-nm lit cells. This is indicative that caspase-3 activation is caused by 254-nm and 277-nm wavelengths as mentioned in LINE 92-94.

  1. Figure 3: the background of the images (3A:222nm, 3D: 254nm) in the red channel should be unified or all compared images. Did the authors consider the background intensity during the measurement/analysis of images?

Response: In the analysis of images for Figure 3, we probed the levels of É£H2AX and thymine dimers by calculating the average intensity of the red channel minus the image background Ibg. Ibg is defined as the average intensity of an image from blank areas where no DAPI signal is observed. Spots that exceeded the threshold levels were then considered as being stained positive. In the revised manuscript, we have since added this information in the Methods under “4.7 Immunofluorescence experiments”.

  1. How many microscopic fields and how many cells were analyzed during the immunofluorescence analysis?

Response: For immunofluorescence analysis, the number of experiments has been indicated in the Figure 3 legends, where n=5 for ɣH2AX imaging and n=3-4 for thymine dimer imaging. Typically for each condition in every experiment, we took a total of 5 microscopic fields, which gives an average of 2000 cells.

  1. Materials and methods: the table is too detailed. I suggest, that it should be presented as supplementary material.

Response: We agree with the Reviewer and have since moved the Key Resources Table to Table S1.

  1. The Authors included the results in the Introduction section. It is not a mistake but I suggest the fragment line 51-line 58 be rewritten.

Response: While we agree with the Reviewer that the paragraph spanning from line 51 to line 58 essentially contains key summary of results in the paper, it is mentioned in the journal format for the section of introduction to finally “briefly mention the main aim of the work and highlight the principal conclusions”. It is therefore justified for the original paragraph to stay as it is.

  1. line 28 – „eye damages” – the term should be explained by including examples.

Response: We have expanded the term “eye damages” to more specific instances in the form of “damages in the cornea and lens” and added the respective references.

  1. The manuscript has many editorial errors, e.g. the caption of each figure requires a complete rework.

Response: We note the Reviewer’s comment and have since made the appropriate changes in the latest manuscript.

  1. Reference list was not prepared according to the Journal guidelines.

Response: We note the Reviewer’s comment and have since made the appropriate changes in the latest manuscript.

Reviewer 2 Report

The manuscript "222-nm far UVC exposure results in DNA damage and transcriptional changes to mammalian cells" (Manuscript ID: ijms-1847078is a comprehensive study that compares the different UVC wavelengths and their effects on cell viability and DNA damage. Furthermore, results indicated that the introduction of 222-nm far UVC lamps for disinfection should be carefully considered and designed with the inherent dangers involved. 

The study appears to be performed with care. The experiments were adequately performed and presented, and the results were interpreted appropriately. Therefore, this manuscript is interesting, scientifically justified, well written and detailed. In addition, the literature is up to date and thoroughly investigated.

The manuscript generally represents original, complete, well-planned and methodologically sound research. 

Some minor typographical and grammatical errors in the manuscript should be checked and corrected. 

Author Response

The manuscript "222-nm far UVC exposure results in DNA damage and transcriptional changes to mammalian cells" (Manuscript ID: ijms-1847078) is a comprehensive study that compares the different UVC wavelengths and their effects on cell viability and DNA damage. Furthermore, results indicated that the introduction of 222-nm far UVC lamps for disinfection should be carefully considered and designed with the inherent dangers involved. 

The study appears to be performed with care. The experiments were adequately performed and presented, and the results were interpreted appropriately. Therefore, this manuscript is interesting, scientifically justified, well written and detailed. In addition, the literature is up to date and thoroughly investigated.

The manuscript generally represents original, complete, well-planned and methodologically sound research. 

Some minor typographical and grammatical errors in the manuscript should be checked and corrected. 

Response: We have made amendments in terms of format, typographical and grammatical errors in the latest manuscript as attached. We thank the Reviewer for the positive comments made.

Round 2

Reviewer 1 Report

The Authors improved the manuscript according to the comments. I recommend it for publication.

Author Response

We thank the Reviewer for his/her kind comments. We have checked the manuscript for minor grammatical errors/spellchecks.